# *Vibrio cholerae* Chromosome Partitioning without Polar Anchoring by HubP

**DOI:** 10.3390/genes13050877

**Published:** 2022-05-13

**Authors:** Christophe Possoz, Yoshiharu Yamaichi, Elisa Galli, Jean-Luc Ferat, Francois-Xavier Barre

**Affiliations:** Université Paris-Saclay, CEA, CNRS, Institute for Integrative Biology of the Cell (I2BC), 91198 Gif-sur-Yvette, France; yoshiharu.yamaichi@i2bc.paris-saclay.fr (Y.Y.); elisa.galli@i2bc.paris-saclay.fr (E.G.); jean-luc.ferat@i2bc.paris-saclay.fr (J.-L.F.); francois-xavier.barre@i2bc.paris-saclay.fr (F.-X.B.)

**Keywords:** partition system, chromosome segregation, chromosome organization, HubP, ParABS, *Vibrio cholerae*

## Abstract

Partition systems are widespread among bacterial chromosomes. They are composed of two effectors, ParA and ParB, and cis acting sites, *parS*, located close to the replication origin of the chromosome (*oriC*). ParABS participate in chromosome segregation, at least in part because they serve to properly position sister copies of *oriC*. A fourth element, located at cell poles, is also involved in some cases, such as HubP for the ParABS1 system of *Vibrio cholerae* chromosome 1 (ch1). The polar anchoring of *oriC* of ch1 (*oriC1*) is lost when HubP or ParABS1 are inactivated. Here, we report that in the absence of HubP, ParABS1 actively maintains *oriC1* at mid-cell, leading to the subcellular separation of the two ch1 replication arms. We further show that *parS1* sites ectopically inserted in chromosome 2 (ch2) stabilize the inheritance of this replicon in the absence of its endogenous partition system, even without HubP. We also observe the positioning interference between *oriC1* and *oriC* of ch2 regions when their positionings are both driven by ParABS1. Altogether, these data indicate that ParABS1 remains functional in the absence of HubP, which raises questions about the role of the polar anchoring of *oriC1* in the cell cycle.

## 1. Introduction

Partition systems have been initially discovered on low-copy-number plasmids. They promote plasmid stability over vertical transmission via partitioning: the spatial distribution of the plasmid molecules along the long axis of rod-shaped cells. Among the different types of partition systems described, Type I systems are preponderant and are the only ones to be found on chromosomes [1,2]. They are composed of an ATPase, ParA, a DNA binding protein, ParB and a cis-acting *parS* DNA motif where cognate ParB specifically binds. They are very conserved, and systematic sequencing of bacterial genomes revealed that they are present on most bacterial chromosomes [3,4]. A cluster of *parS* sites identifying the DNA molecule to be partitioned is usually located near the replication origin of the chromosome, *oriC* [4]. As *parS* sites are the first locus to be segregated, the coincidence between *parS* and *oriC* loci is crucial to segregate in the replication order [5,6].

The mechanism of Type I partitioning systems, studied first on plasmids and then on chromosomes, is thought to function via a Brownian ratchet model (reconstituted in vitro [7] and reviewed in Refs [2,8]). Upon the initial specific interaction with individual *parS* sites, ParB spreads onto adjacent DNA segments [9,10,11,12]. A recent breakthrough revealed that the ParB family of proteins has emerged as a class of molecular switches that require CTP for proper function [13,14,15,16]. CTP induces ParB self-dimerization to create a clamp-like molecule. The ParB clamp self-loads at *parS*, then spreads by sliding to neighboring DNA while still entrapping DNA [2,14], resulting in the formation of a higher-order nucleoprotein complex or “cargo” with multiple ParB-CTP clamps entrapped in the 10 to 20 kb vicinity of the *parS* locus. ParA binds DNA non-specifically when complexed with ATP (ParA^ATP^). The ParB cargo interacts with DNA-bound-ParA^ATP^ and catalyzes the hydrolysis of its ATP. ParA^ADP^ is then released from the nucleoid and remains in the cytoplasm until ATP replaces ADP. Thus, the ParB cargo progresses on the ParA^ATP^-covered nucleoid via transient interactions without any possibility of way back, which drives partitioning.

The final subcellular localization of *oriC* was often reported to be polar [17,18,19,20,21]. In contrast, the final positioning of the ParB cargo in plasmid systems is, on average, at mid-cell when in unique copy and at ¼ and ¾ positions of the cell when in two copies [2,7,22]. The polar localization of *oriC* is dependent on an additional component of the partition system, which is not conserved among bacteria [23,24,25]. In *Vibrio cholerae*, whose genome is divided into two chromosomes (chromosome 1 and chromosome 2, referred to as ch1 and ch2, respectively), each chromosome specifying its own partition system, ParABS1 and ParABS2, respectively. There are a few naturally occurring single-chromosome *V. cholerae* strains that are an exception to the two-chromosome rule [26]. However, these single chromosomes originate from the fusion of the two chromosomes, and the concertation between the two partition systems has not been studied. Even though both belong to Type I partitioning system, ParABS1 is phylogenetically close to the chromosomal group, and a unique cluster of three *parS1* sites is located about 60 kb from *oriC1*. ParAB2 system, in contrast, groups with plasmid-type partition systems [27], and *Vibrio*-specific *parS2* sites are more numerous and scattered over ch2 [3,28]. The polarly localized protein HubP shown to titrate ParA1^ADP^ is responsible for the directional movement of one sister *oriC1* from a pole to the opposite pole and for the *oriC1* polar anchorage [29]. In *V. cholerae*, the absence of HubP or any of the other three components abolishes the polar localization of *oriC1* but does not impede chromosome segregation [6,28,29,30]. In contrast, ParABS2 is not HubP anchored and is required for the proper inheritance of the 1 Mbp-sized replicon [31]. Similarly, PopZ in *Caulobacter crescentus* and DivIVA in *Corynebacterium glumaticium* are required for *oriC* polar anchoring [23,24]. In *C. crescentus*, the depletion of PopZ, as the absence of the other three components, is lethal due to chromosome and cell division segregation defects [23,32]. 

In *V. cholerae*, *oriC1* positioning was reported to be similar in *hubP*- and in *parAB1*-deleted strains [29]. However, a possible modification of the ParA1 recycling rate due to the absence of HubP could affect the motion of ParB cargos [33]. In this manuscript, we investigate whether the partition system of ch1 still presents some partition activity in the absence of the HubP element. First, we observed that in the heterologous host *Escherichia coli*, the *V. cholerae* ParABS1 system did not fully stabilize a low copy-number plasmid. However, we found that ParABS1 promotes the efficient stabilization and localization of *parAB2*-deleted ch2 in Δ*hubP V. cholerae.* Moreover, we observed an interference of positioning between *oriC1* and *oriC2* regions when both regions contain *parS1* sites. Importantly, we revealed that in the absence of HubP, the ParABS1 system continues to act on its natural target *oriC1* region. While released from the pole, *oriC1* is actively maintained at mid-cell. This action also triggered a global rearrangement of ch1. 

## 2. Materials and Methods

### 2.1. Plasmids and Strains

The bacterial strains and plasmids used in this study are listed in Appendix A, respectively. All *V. cholerae* strains are derivatives of the El Tor N16961 strain rendered competent by the insertion of *hapR* by specific transposition and constructed by natural transformation [34]. Engineered strains were confirmed by PCR. 

### 2.2. Construction of MiniF-Plasmid Derivatives

Oligo DNAs containing the *parS1* sequence (YPR179 and YPR180) were annealed and cloned into the EcoRI site of the mini-F plasmid pXX705 [35], resulting in pYB164. parAB1 genes were amplified with YPR254 and YPR255 (Appendix A) followed by TOPO cloning into the pCR Blunt II-TOPO vector (ThermoFisher), resulting in pYB145. The parAB1 fragment was then excised by BamHI digestion and cloned into the same site of mini-F plasmids. Nucleotide sequence was confirmed by Sanger sequencing.

### 2.3. Construction of Plasmids Used for Strain Constructions

In order to replace or insert elements at a precise position, both the 1-kb upstream DNA segment amplified with the UP oligos and digested *Xba*I/*Xho*I, and the 1-kb downstream DNA segment amplified with the DW oligos and digested *Bam*HI/*Sac*I were cloned in an R6K *E. coli* vector. A resistance marker was then cloned in between. The oligos used for pPOS209 (to delete *parAB2*) and pPOS188 (to delete the 3 *parS1* region) are listed in Appendix A.

### 2.4. Plasmid Stability Assay

The plasmid stability assay was carried out as previously described [27]. Essentially, *E. coli* harboring mini-F plasmid were grown without ampicillin. Cultures were back-diluted to keep the cells in the log phase. At time points, cells were spread on LB plates (without ampicillin) to form colonies. The fraction of plasmid-retaining cells was measured by patching 200 colonies to the LB plate containing 25 µg/mL ampicillin.

### 2.5. Fluorescence Microscopy

Cells were grown in M9 minimal medium supplemented with 0.4% fructose and 1 µg mL^−1^ thiamine to the exponential phase and spread on a 1% (weight/vol) agar pad (ultrapure agarose, Invitrogen) for analysis. For snapshot analyses, at least 500 cell images were acquired using a DM6000-B (Leica) microscope, and they were analyzed using MicrobeTracker [36]. For time-lapse analyses, the slides were incubated at 30 °C, and images were acquired using an Evolve 512 electron-multiplying charge-coupled device (EMCCD) camera (Roper Scientific) attached to an Axio Observe spinning disk (Zeiss). At each time point, we took a stack of 32 bright-field images covering positions 1.6 μm below and above the focal plane. Cell contours were detected, and cell genealogies were retraced with a MatLab-based script developed in the laboratory [37]. After the first division event, the new pole and old pole of cells could be unambiguously attributed based on the previous division events.

## 3. Results

### 3.1. ParABS1 System Does Not Efficiently Stabilize a Mini-F in E. coli

The activity of a partition system can be tested in *E. coli* by monitoring the stability of a mini-F plasmid lacking its own partition system SopABC, hereafter referred to as miniF. This heterologous host assay has been used previously to demonstrate the stabilization capacity of several plasmid and even chromosome partition systems, such as those of *Bacillus subtilis*, *Pseudomonas putida*, *Burkhlderia cenocepacia* [17,27,38] and *V. cholerae* ch1 [30]. However, the level of stabilization can be affected by the level of ParAB expression. For instance, the stabilization of miniF by the *B. cenocepacia* chromosome 1 partition system was only obtained in conditions of overexpression of the *B. cenocepacia* ParAB proteins [17]. Likewise, Saint-Dic et al. reported that the *V. cholerae* partition system stabilized a miniF harboring a *parS1* site when the *V. cholerae* ParA1 and ParB1 were expressed from a p*lac* promoter on a multicopy vector, i.e., in conditions of overproduction. Therefore, it remained to be tested whether ParA1 and ParB1 could stabilize a miniF harboring a *parS1* site when produced under more physiological levels. To this end, we introduced the *parAB1* operon under its endogenous promoter and a *parS1* site in a miniF (miniF-*parABS1*).

The stability (i.e., the vertical inheritance of the plasmid among cell population across generations) of the miniF-*parABS1* plasmid was assessed over ≈20 generations. The ratio of cells harboring the plasmid was assessed at different time points of cultivation without selective pressure (Figure 1). Alongside, we used, as a positive control, a miniF plasmid carrying *parABS2*, the partition system of *V. cholerae* ch2 (miniF-*parABS2*), known to be independent of HubP and active in *E. coli* [28]. As negative controls, we analyzed the loss of the “empty” miniF plasmid and miniF plasmids containing either *parS1* or *parAB1* only. The fraction of cells harboring an unstable plasmid decreases exponentially as a function of the number of generations. Hence, the rate of loss per generation can be estimated as the slope of the linear regression of the logarithm of the number of cells harboring the plasmid as a function of the number of generations (Figure 1). The miniF plasmids lacking part of the system, miniF-*parS1* and miniF-*parAB1,* had a loss rate of 12–13% per generation, similarly to the “empty” miniF plasmid. The miniF-*parABS2* plasmid had a loss rate of only 2% per generation. In contrast, the ParABS1 system only partially restored the stability of the miniF, with a loss rate of 8% per generation. These results suggested that HubP was necessary for the full activity of the ParABS1 system under physiological expression levels of *parA1* and *parB1*.

### 3.2. HubP Is Dispensable for the ParABS1-Driven Stabilization of ch2 in V. cholerae

It was not possible to test whether HubP was necessary for full activity of the ParABS1 system in *E. coli* because HubP does not localize at cell poles in this organism [29]. Therefore, we decided to monitor the partition activity of ParABS1 directly in *V. cholerae*. *V. cholerae* possesses defense mechanisms that impeded the proliferation of plasmids independently of segregation problems [39]. However, we could take advantage of the essentiality of the ParABS2 system of ch2 for its segregation [28]. We created a strain in which a cluster of two *parS1* sites was inserted at the *ori2* locus, 60 kb away from the origin of replication of ch2, and in which the *parAB2* operon was deleted (ch2[Δ*parAB2 ori2*::2*parS1*]). The possibility of creating such a strain demonstrated that the ParABS1 system stabilized ch2[Δ*parAB2 ori2*::2*parS1*]. We analyzed the viability of the strains by a simple drop assay, i.e., for each strain culture (in LB to OD_600 nm_ = 0.2), 10 µL-drops of 10-fold serial dilutions were deposited on LB agar plate and incubated overnight. The size and number of colonies that the strain could form were comparable to a wild-type (WT) strain (Figure 2A). In contrast, a Δ*parAB2* mutant strain could only produce tiny colonies because of the instability of ch2[*parAB2*] (Figure 2A) [31]. Importantly, the deletion of *hubP* did not affect the stability of the ch2[Δ*parAB2 ori2*::2*parS1*] carrying strain to form colonies, demonstrating that HubP is not essential for ParABS1-driven partition. However, between 10 and 20% of the Δ*hubP* Δ*parAB2 ori2*::2*parS1* cells were of a longer length than WT cells, whereas Δ*parAB2 ori2*::2*parS1* cells had a similar length distribution to WT cells (Appendix A).

We equipped the WT, Δ*parAB2 ori2*::2*parS1* and Δ*hubP* Δ*parAB2 ori2*::2*parS1* strains with a dual fluorescent labeling system to detect the number and the positions of the foci of two loci: R2II (located at 120 kb from the ectopic *parS1* sites in the *oriC2* region) and L1I (located at 300 kb from *oriC1*). We observed that the proportion of Δ*parAB2 ori2*::2*parS1* and Δ*hubP* Δ*parAB2 ori2*::2*parS1* mutant cells containing one or two foci of each locus was lower (75% and 50%, respectively) than the proportion of WT cells containing one or two foci (90%) in snapshot images of slow exponentially growing cells (Appendix A).

Taken together, these results showed that the addition of *parS1* on ch2 restored some stability to ch2[Δ*parAB2*] but altered the normal cell cycle progression. The deletion of *hubP* or of the ch1 endogenous *parS1* sites exacerbated these perturbations.

Viability deduced from the cfu counting of the serial dilution (10^−4^ to 10^−8^) of the indicated strains grown in LB to OD_600 nm_ = 0.5 (A). Graph of reconstituted choreographies of *L1I* (red) and *R2II* (green) loci in ADV27 (B), in CP789 (C) and in CP797 (D) from snapshots analysis. Pole 1 was determined using the most polar *L1I* focus as a reference. The median, the 25th and the 75th percentiles of the relative cell position of each locus are plotted for each cell size interval. A schematic of foci choreographies is placed on the right of each graph combining snapshot and timelapse data (Appendix A). In each cell scheme, the old pole and the new pole are represented by a black and white shading, respectively.

### 3.3. ParABS1-Driven Positioning of ch2 in V. cholerae in Absence of HubP

We reconstituted the choreographies of R2II and L1I loci as a function of cell length to determine whether the stabilization of ch2[Δ*parAB2 ori2*::2*parS1*] correlated with an active positioning of ch2 along the long cell axis. We limited our analysis to cells exhibiting at most two foci of either locus. Cells were binned in 0.5 µm intervals from 2 µm to 5.5 µm, the size range containing a sufficient number of cells per bin to be analyzed. The cells were oriented using the most polar L1I focus as a reference. The reconstituted choreographies of L1I in WT and Δ*hubP* Δ*parAB2 ori2*::2*parS1* cells were quite similar; in small-size cells, the unique L1I focus was positioned toward one of the poles and, after duplication, one focus remained at its initial position, while the other migrated toward the opposite cell pole. As the L1I locus is about 300 kb from *oriC1*, it explains that its positioning is not so dependent on HubP. In Δ*parAB2 ori2*::2*parS1* cells, the L1I focus was positioned toward a pole and, after duplication, one focus remained polar and the other migrated to the other cell side toward mid-cell. Hence, after cell division, L1I was located toward the old pole in one daughter cell and toward the new pole in the other daughter cell (Figure 2B–D). The choreography of the R2II locus was strongly modified in ch2[Δ*parAB2 ori2*::2*parS1*] harboring strains. In the smaller Δ*parAB2 ori2*::2*parS1* and Δ*hubP* Δ*parAB2 ori2*::2*parS1* cells, the single R2II focus was located toward the pole opposite to the L1I pole, whereas in WT cells, it was positioned close to mid-cell (Figure 2B). In WT cells, R2II duplication occurred at mid-cell, and the two sister foci relocated at the ¼ and ¾ positions (Figure 2B). In Δ*hubP* Δ*parAB2 ori2*::2*parS1* cells, the R2II focus migrated toward mid-cell prior to duplication, and the two sister foci remained close to mid-cell until cell division (Figure 2D). In Δ*parAB2 ori2*::2*parS1* cells, R2II duplication occurred at the pole, with one of the two sister foci relocating toward mid-cell on the opposite cell side. Hence, *oriC1* and *oriC2* proximal regions exhibited mirror choreographies (Figure 2C).

Each cell contains an old pole inherited from its mother and a new pole resulting from the latest binary scission. Timelapse experiments showed that the single L3I focus of the smaller WT and Δ*hubP* Δ*parAB2 ori2*::2*parS1* cells, i.e., the newborn cells, was positioned at the old pole (Ref [6] and Appendix A). In contrast, the single L3I focus of the smaller Δ*parAB2 ori2*::2*parS1* cells was located toward either the old or the new cell pole (Appendix A). Taking into account snapshot and timelapse data, we could reconstitute the choreographies of L3I and R2II in the different strains (cell schemes of Figure 2B–D).

Taken together, these data showed that the ParABS1-mediated stabilization of ch2 was obtained through active positioning of the *oriC2* region. Moreover, we observed a positioning interference between the two *parS1*-containing replicons that cannot colocalize. Correspondingly, deleting the *parS1* sites on ch1 in the Δ*parAB2 ori2*::2*parS1* strain led to different L1I and R2II choreographies (Appendix A).

### 3.4. ParABS1 System Positions oriC1 in V. cholerae in Absence of HubP

As the ParABS1 system showed some partition activity on ch2 in the absence of HubP, we decided to revisit the positioning of the *oriC1* region in the absence of HubP, relative to another locus of ch1. To this end, we analyzed in a WT strain (ADV24), a *hubP*-deleted strain (CP700) and a 3*parS1*-deleted strain (ADV40), the choreography of a locus located at 15 kb from the endogenous *parS1* sites, ori1, and of a locus located 650 kb away from the *parS1* sites, L3I*,* both on the left replication arm. Cells ranging from 2.2 to 4.6 µm were binned in 0.2 µm intervals. *hubP* mutant cells were slightly longer than WT or Δ3*parS1* cells (Appendix A). In the WT and *parS1*-deleted strains, ori1 foci were more polar than L3I foci (Ref [6] and Appendix A). Therefore, we could reconstitute the choreographies of ori1 and L3I based on single snapshot images using the most polar ori1 focus as a reference (Figure 3A,B). The WT and *parS1*-deleted strains showed similar behavior, ori1 foci being polar, while a single L3I focus was located at mid-cell, and sister L3I foci were located at the ¼ and ¾ positions. However, we noted that ori1 foci were closer to the poles in WT cells than in Δ*parS1* cells (Figure 3A,B). In contrast, video microscopy revealed that ori1 was less polar than L3I in the *hubP*-deleted strain (Appendix A). Indeed, averaging 56 independent cell lineages showed that the single ori1 focus of newborn cells was positioned at mid-cell and that sister ori1 foci migrated at the ¼ and ¾ positions after duplication (Figure 3C). Then, we reconstituted the choreographies of ori1 and L3I foci in the *hubP*-deleted strain based on single snapshot images using the most polar L3I focus as a reference. It confirmed the mid-cell positioning of ori1 (Figure 3D). The interfocal distance, among shorter cells (<3 µm), between ori1 and L3I in the *hubP*-deleted strain was even more reduced than in the 3*parS1*-deleted strain, confirming the modification of the chromosome organization (Figure 3E). In conclusion, the non-anchored ParABS1 system mediated the sharp positioning of *oriC1* at mid-cell before replication and that of *oriC1* at ¼ and ¾ of the cell, after duplication, like the classical plasmid partition system.

### 3.5. Chromosome I Rearrangement by Non-Anchored Partition System

In order to deduce the global arrangement of ch1 within the cells, we determined by snapshots analysis the positioning of L3I in combination with R2I, a locus on the other replication arm. Cells were arbitrarily oriented using the most polar R2I focus as a reference. As previously observed, L3I and R2I foci colocalized during the entire cell cycle in the WT strain (Figure 4A) [27]. There was slightly less colocalization in the Δ*parS1* strain (Figure 4B) [6]. In contrast, the L3I-R2I colocalization was lost in the Δ*hubP* strain (Figure 4C), which led to an increase in the interfocal distance between L3I and R2I foci in short cells (<3 µm) (Figure 4D). Video microscopy further revealed a strong heterogeneity of the choreography of L3I foci in the different cells (Figure 4E). Notably, in newborn cells, in which the orientation is directly tracked from the division of the mother cell, the L3I focus was positioned in about similar proportions toward the old (27/56, Appendix A) or toward the new pole (21/56 Appendix A). Averaging non-homogenous lineages resulted in a blurry and uninformative choreography (Figure 4F).

Taken together, these data suggested that only *oriC1* followed a defined choreography from mid-cell toward the ¼ and ¾ positions, i.e., the mid-cell of the two future daughter cells, whereas the mere constraint on L3I and R2I loci would be to locate on opposite sides of *oriC1* (Figure 4G). Interpolating the genomic region spanning L3I-ori1-R2I, our results suggest that in the absence of HubP, but not *parS1*, the longitudinal arrangement of WT ch1 is switched to a lateral arrangement with each replication arms on separate cell halves. As a corollary, non-anchored ParABS1 would contribute to the global rearrangement of ch1 by positioning *ori1* at mid-cell (Figure 4H).

## 4. Discussion

Compared to the role of the ParABS system in the proper inheritance of low-copy-number plasmids, its role in chromosome segregation is more complex and has been a question of debate [2]. Indeed, the phenotypes associated with their inactivation in different species varied from lethal to dispensable. Another difficulty came from the pleiotropic roles played by the partition systems. The ParABS systems were found to participate in the regulation of gene expression [40], chromosome replication initiation [41,42] and in cell division licensing [43]. Therefore, it is difficult to determine which part of the *par* phenotype could be attributed to loss of partitioning.

Here, we investigated whether HubP, the polar fourth component of *V. cholerae* chromosome I partition system, might be considered a critical contributor of its partition activity. Therefore, we explored to what extent a chromosomal ParABS system could still be functional in the absence of anchorage. Our data unambiguously demonstrated in both natural and heterologous hosts (*E. coli*) that the ParABS1 system of *V. cholerae* continues to exhibit partitioning activity in the absence of HubP. It was observed in the stabilization and the active positioning of the Δ*parAB2*-ch2 and its endogenous target, the *oriC1* region. The loss of polar positioning of *oriC1* in both 3*parS1* and *hubP* mutants was already reported but never analyzed side by side, which explains why the difference between their phenotypes was not revealed earlier.

### 4.1. Intrinsic Adaptability of the Partition Systems

This report also revealed the intrinsic adaptability of the partition system to function in different environments, i.e., with or without anchoring (*hubP* mutant), and with twice more replicons to distribute (for instance, when *parS1* sites were present both on ch1 and on ch2). In the absence of HubP, the ParABS1 system of ch1 appeared to act as the *parABS2* of ch2 or as any Type I-plasmid partition systems.

The role of HubP is presumably to tether the inactive form of ParA1 at the pole, which should affect the localization and the recycling rate of ParA1. However, this role appeared to be not an essential element of the partition activity. As ParA1 in *hubP* mutant cells is free to be reactivated anywhere in the cell, the nucleoid could be constantly recoated with active ParA1. The ParB1-*parS1* complex would then be trapped in highly dense regions of the nucleoid (HDR), as proposed for plasmid systems [44]. Moreover, a modification of the recycling rate of a plasmid ParA was shown to affect but not abolish its partition activity [45]. HubP polar tethering of inactive ParA1 probably also acts as HDR to trap the ParBS1 complex, and the recycling rate of ParA1 between these two trapping systems might not be sufficiently different to have any impact on the partition activity.

### 4.2. Positioning Interference of oriC1 and oriC2, the Two parS1-Containing Regions

We showed that the ParABS1 could actively position ch2[Δ*parAB2 ori2*::2*parS1*] along the long cell axis. However, this positioning differed depending on whether ch1 contained *parS1* sites or whether the strain expressed HubP. When only ch2 contained *parS1* sites (CP799, Appendix A), the *oriC2* region behaved as *oriC1*, located at the old pole in newborn cell. When both chromosomes contained *parS1* sites (CP789, Figure 2C), the *oriC1* and *oriC2* regions were located at opposite poles because they might be recognized as two sister copies of *oriC1* regions and positioned at the opposite pole. After *oriC1* duplication, one sister *oriC1* migrated from its pole toward the other cell half, but the progression could be stopped by the presence of the *parS1* containing *oriC2* regions anchored by HubP at the opposite pole. Hence, this sister *oriC1* would position at mid-cell, at mid-distance between the two “*parS1*-occupied” poles. Then, after *oriC2* duplication, one sister *oriC2* would be mobilized from the pole toward the other cell half. The mechanism allowing the proper arrangement of *oriC1* and *oriC2* remains elusive. We could speculate that positioning two *oriC2* in one cell half and two *oriC1* in the other could be too unbalanced, as ch1 is three times the size of ch2. Hence, in most of the cases, it appeared that the arrangement was *oriC1*-*oriC2*-*oriC1*-*oriC2*, and cell division generated two daughter cells with either *oriC1* or *oriC2* at the old pole. In contrast, in the absence of HubP (CP797, Figure 2D), we observed that newborn cells systematically contained *oriC1* at the old pole. This could be due to the displacement of *oriC2* by the migration of one sister *oriC1* because *oriC2* was not anchored by HubP. Then, *oriC2* would position in between the two *oriC1* and duplicate at mid-cell.

### 4.3. Loss of ch1-Anchoring Leads to Transversal Organization of ch1

Most bacterial chromosomes are longitudinally arranged. In slow-growing newborn cells, this arrangement is characterized by the positioning of their *oriC* region toward the old pole (not necessarily attached to it), the positioning of the replication terminus region (*ter*) toward the new pole and the juxtaposition of the two replication arms along the long cell axis (as illustrated for the WT in Figure 4H). The <*oriC*-*ter*> arrangement does not rely primordially on the replication program but on the proximity between *oriC* and *parS* loci, as the *parS* cluster drives the global arrangement independently of its genomic position [5,6]. However, *parS1* insertions are not similarly tolerated at all genomic positions [46,47]. In *V. cholerae* ParABS1 mutants, the anchoring of *oriC1* was lost, and the positioning of any loci, except in the ter region, was loose [6]. Yet, the arrangement of ch1 remained longitudinal, not because of HubP action but because the earlier a locus is replicated, the farther it is segregated, as was proposed in Ref [6] (Figure 4H). In *hubP* mutants, the partition activity positioned *oriC* at mid-cell and, in doing so, unexpectedly triggered the global re-arrangement of ch1. The two replication arms now occupied separated cell halves corresponding to a transversal organization. This organization is reminiscent to that of *E. coli*, naturally devoid of a *par* system [48,49]. In *E. coli*, the transversal organization relies on the action of the SMC-like protein complex, MukBEF [50,51,52,53]. MukBEF action mechanism is proposed to be the lengthwise compaction of the full chromosome, except for the *ter* region, which is protected from it by the MatP protein [51,54]. MukBEF is also present in *V. cholerae*, but here, we showed that the transversal arrangement of ch1 also required the polar release and ParABS1 activity. As MukB and ParABS1 systems might cooperate in ch1 segregation, the “required” activity of *Vc*-MukB to “segregate” ch1 and ch2 could be weaker than that of the *Ec*-MukB one to segregate its *par*-less chromosome. The heterologous expression of *Ec*-MukB in *P. aeruginosa* is compatible with this possibility, as it conferred an enhanced complementation [55]. Nevertheless, other parameters, such as the timing order of ch1 and ch2 replication and/or the chromosomes size difference, could be required to trigger a transversal switch of chromosome arrangement, as ch2 is longitudinally arranged.

### 4.4. Role of the Polar Anchoring of the Origin of Replication

It is probable that in *V. cholerae*, as reported in different bacteria, such as *P. aeruginosa* [56] or *B. subtillus* [57], the SMC-type complexes and the ParABS system have a redundant action on segregation. It would be interesting to analyze the phenotype of the *mukB parAB1* and *mukB hubP* double mutants in *V. cholerae*. If the viability of these two double mutants is reduced, it could indicate that the partitioning of *oriC1* contributes to chromosome segregation but that the anchoring is required for full efficiency of the process. This, in turn, could be problematic for the segregation of ch2. We observed cell cycle perturbations in cells in which ParABS1 mediated the positioning of ch2 (Appendix A, CP789, CP799). Mid-cell positioning of sister *ter* regions of ch2 (ter2) participates in proper cell division licensing [58]. The excessive separation of the sister *ter2* due to *oriC2* polar anchoring could be responsible for the cell cycle perturbations. We observed that the duplication of R2II in CP789, CP797 and CP799 occurred at a larger cell mass (Figure 2C,D and Appendix A), which might be a consequence of these cell cycle perturbations.

In addition to the possible role in chromosome segregation enhancement, the polar positioning of *oriC1* region toward the old pole could also have a role in enhancing the proper sublocalization of certain proteins acting within the polar area. Additionally, the arrival of *oriC1* region to the new pole could contribute to the efficient division licensing, as in *C. crescentus* [43]. In *V. cholerae*, FtsZ protein, which is a scaffold for recruiting cell division machinery, remained after cell division site, i.e., new pole in progeny cell [58]. Previous work showed that it is linked to a cell division inhibitor, SlmA, which binds to specific sites along the two *V. cholerae* chromosomes, with the exclusion of the *ter* domains [57]. Indeed, due to the longitudinal arrangement of ch1 from *oriC1* at the old pole to the ter regions at the new pole, DNA-bound SlmA is rare in the new pole area until newly replicated copy of oriC1 is segregated. Then, the ParABS1 system could have a role in the timing of cell division by bringing one of the two sister SlmA-bound *oriC1* regions toward the new cell pole, which would displace FtsZ and trigger the reassembly of the FtsZ ring at the future division site.

Further work analyzing timing and positioning at a more resolutive level than our present study will be required to demonstrate whether the polar positioning of *oriC1* could play any of those proposed roles in chromosome segregation, protein localization and cell division licensing.

## Figures and Tables

**Figure 1 genes-13-00877-f001:**
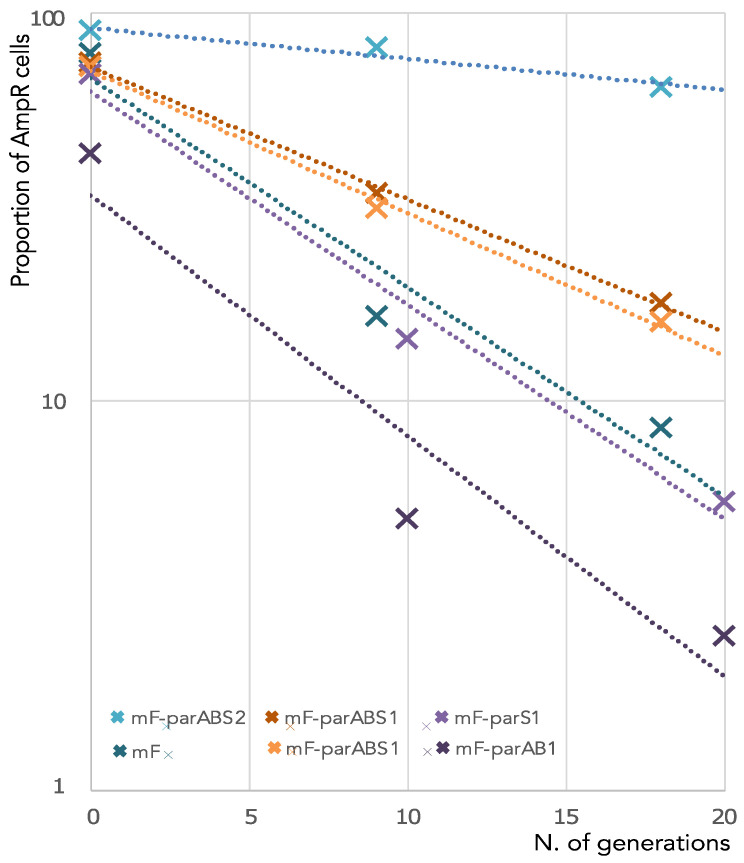
Stability of miniF plasmid derivatives in *E. coli*. The graph represents the proportion (in %) of Amp^R^ cells (containing plasmid) at different time points during ≈20 generations of LB growth of an *E. coli* strain containing one of the different miniF derivatives indicated in the figure inset. Regressions of the data are shown by lines with the same color codes as data, and their slopes correspond to the proportion (%) of plasmid loss per generation.

**Figure 2 genes-13-00877-f002:**
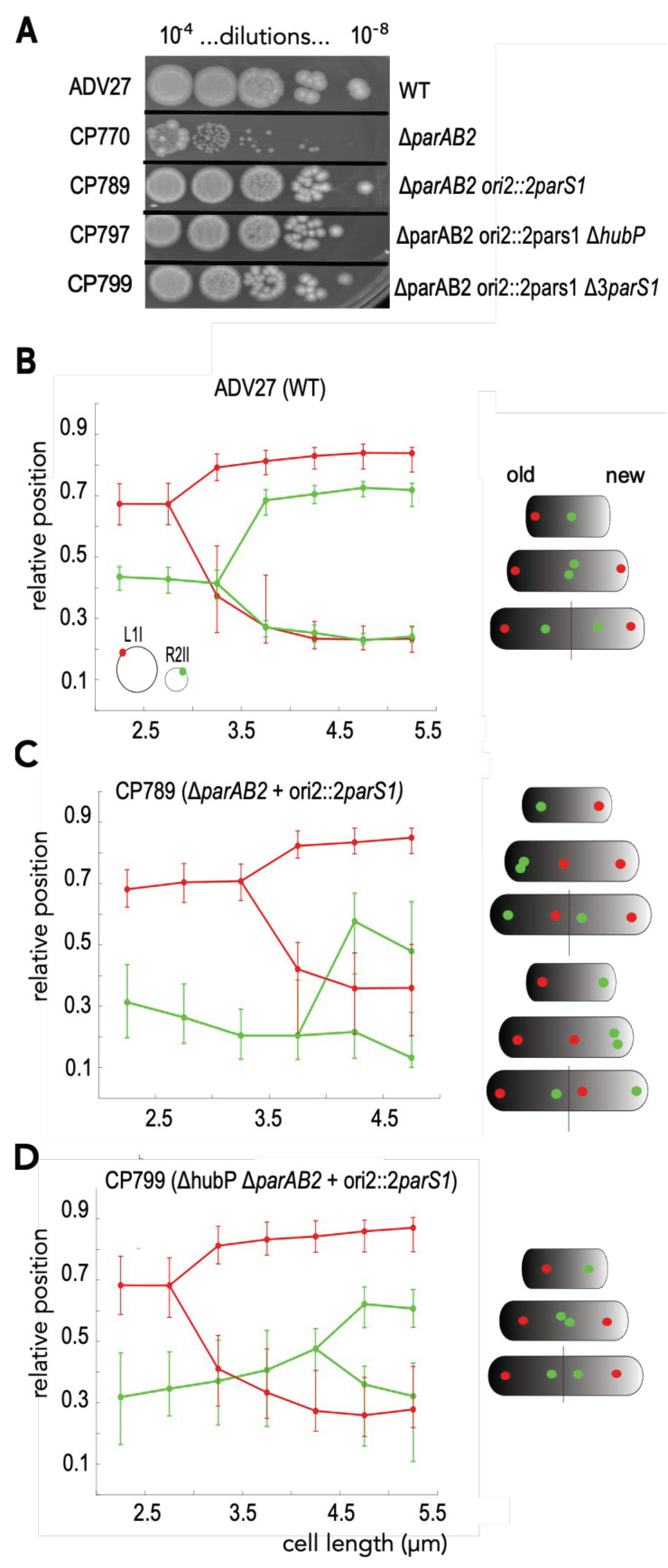
Stabilization and positioning of ch2−Δ*parAB2* by ectopic *parS1* near *oriC2*. Viability deduced from the cfu counting of the serial dilution (10^−4^ to 10^−8^) of the indicated strains grown in LB to OD_600 nm_ = 0.5 (**A**). Graph of reconstituted choreographies of L1I (red) and R2II (green) loci in ADV27 (**B**), in CP789 (**C**) and in CP797 (**D**) from snapshots analysis. Pole 1 was determined using the most polar L1I focus as a reference. The median, the 25th and the 75th percentiles of the relative cell position of each locus are plotted for each cell size interval. A schematic of foci choreographies is placed on the right of each graph combining snapshot and timelapse data (Appendix A). In each cell scheme, the old pole and the new pole are represented by a black and white shading, respectively.

**Figure 3 genes-13-00877-f003:**
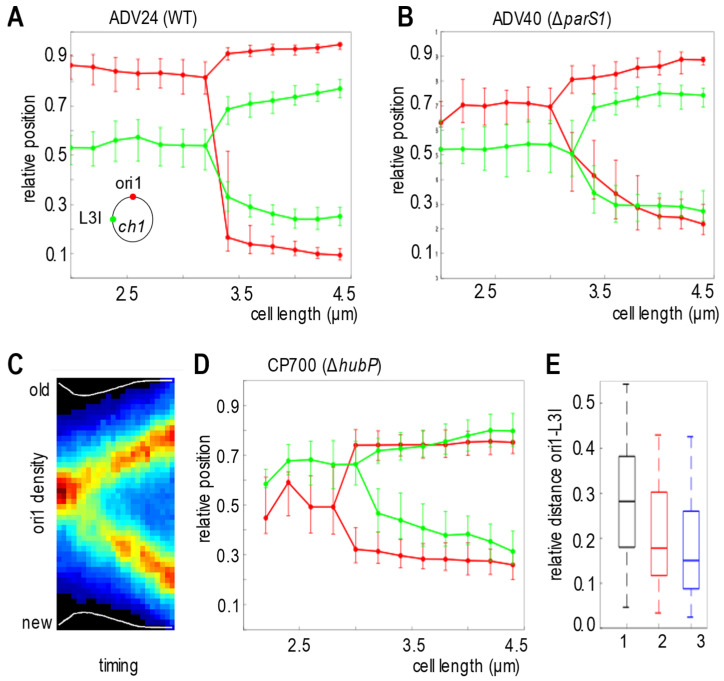
Mid-cell positioning of ori1 in Δ*hubP* cells. (**A**,**B**,**D**) Reconstituted choreographies of ori1 and L3I foci in ADV24 (WT) (**A**), ADV40 (Δ3*parS1*) (**B**) and CP700 (Δ*hubP*) (**D**) from snapshots analysis. In (**A**,**B**), pole 1 is determined using the most polar ori1 focus as a reference (Ref [6] and Appendix A). In (**D**) pole 1 is determined using the most polar L3I focus as a reference (Appendix A). The median, the 25th and the 75th percentiles of the relative cell position of each locus are plotted for each cell size interval of 0.2 µm. (**C**) Average positioning of ori1 obtained from the compilation of 56 lineages of Δ *hubP ori1*-tagged cells videotracked over an entire cell cycle. In the heat map, black corresponds to the lowest and dark red to the highest ori1 fluorescence intensities. y axis: position along the cell length, with 0 corresponding to the new pole and 1 to the old pole. x axis: cell cycle, with 0 corresponding to birth and 1 to scission. (**E**) The relative distance between ori1 and L3I loci was measured as a function of the relative cell length in the cells containing only one focus of each locus. The median (horizontal bar), the 25th and the 75th percentiles (open box) and the 5th and the 95th percentiles (error bars) of the inter-foci distance were indicated for strain 1 (ADV24), 2 (ADV40) and 3 (CP700).

**Figure 4 genes-13-00877-f004:**
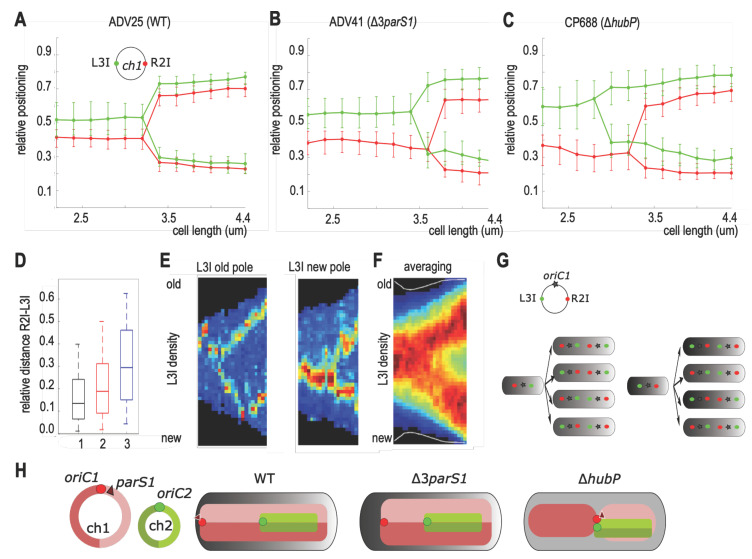
Global ch1 rearrangement in Δ*hubP* cells. (**A**–**C**) Reconstituted choreographies of L3I and R2I foci in ADV25 (WT) (**A**), ADV41 (Δ*parS1*) (**B**) and CP688 (Δ*hubP*) (**C**) from snapshots analysis. Pole 1 was arbitrarily chosen the closest to a R2I focus. The median, the 25th and the 75th percentiles of the relative cell position of each locus are plotted for each cell size interval. (**D**) The relative interfocal distance between L3I and R2I loci was measured as a function of the relative cell length in the cells containing only one focus of each locus. The median (horizontal bar), the 25th and the 75th percentiles (open box) and the 5th and the 95th percentiles (error bars) of the inter-foci distance were indicated for strain 1 (ADV25), 2 (ADV41) and 3 (CP688). (**E**) Two lineage examples of Δ*hubP* L3I videotracking illustrating the heterogeneity of the behavior of this locus: in the newborn cell, L3I was located either toward the old pole (left) or the new pole (right) and in the dividing cell at various final positions. (**F**) Average positioning of L3I obtained from compiling 56 lineages of L3I tagged cells videotracked over an entire cell cycle. In the heat map, black corresponds to the lowest and dark red to the highest L3I fluorescence intensities. y axis: position along the cell length, with 0 corresponding to the new pole and 1 to the old pole. x axis: cell cycle, with 0 corresponding to birth and 1 to scission. (**G**) Schematic model of the eight possible choreographies of L3I-ori1-R2I foci explaining the blurry choreography obtained in (**F**). (**H**) Circular map of ch1 (red) and ch2 (green) (left). Arrangement of ch1 (red) in WT, Δ*parS1* and Δ*hubP*. Ch2 arrangement in Δ*hubP* was not studied and is only assumed.

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
