# Peer review of "Vibrio cholerae Chromosome Partitioning without Polar Anchoring by HubP"

_genes, 2022, doi:10.3390/genes13050877_

Round 1

Reviewer 1 Report

Partition systems are widespread among bacterial chromosomes. Vibrio cholerae is an exception to the  the vast majority of bacteria as its genome partitioned between two unequally sized chromosomes. This unusual two-chromosome arrangement in V. cholerae has sparked considerable research interest since its discovery. The manuscript entitled "Vibrio cholerae Chromosome Partitioning without Polar An- 2 choring by HubP" is an interesting read.  Authors observed positioning interference between oriC1 and oriC2 regions when their positioning are both driven by ParABS1, which improved our understanding about the role of the polar anchoring of oriC1 in the cell cycle. The study is well-designed. However, there is a bit of clarifications need to be done to link the data with interpretations.

1. Introduction: the introduction section is well organized. However, there is a few of naturally occurring single-chromosome V. cholerae that are an exception to the two chromosome rules (etc. doi: 10.1155/2017/8724304). Thus, it worths mentioned in the introduction as well as discussion section.
2. Are 10 generations enough for plasmid stability in E.coli?
3. line 139: "cell harboring the plasmid was assessed at different time", two spaces between "assessed"  and "at". Also checked throughout the manuscript.
4. Figure S7 and S8 are a bit of confusing. A supplymentary movie is much appreciated.
5. line 202: "We limited our analysis to cells with a cell length ranging from 2 µm to 5.5 µm" The reasons need to be provided.
6. line 393:"The ParABS1 system could have a role in the timing of cell division by bringing one of the two sister SlmA-bound oriC1 regions towards the new cell pole". More discussion are needed for this point.
7. line 399: the limitation of this study need to be mention.

Reviewer 2 Report

My general impression on the manuscript is good. I find the manuscript interesting and presenting important results. The results and discussion are a bit hard to follow, I would encourage the authors to delicately simplify their message. As I understand, the quality of the figures is not final (for now some of them are not acceptable). I have several comments to the manuscript, which could be considerated to improve it and make it easier to follow for the readers:

  • I would suggest to exchange the Fig 1. with Fig S1., because I find the results presented as in the Fig S1. more understandable. It would be also possible to present it as in Fig 1., but not as the loss rate, but rather the retention rate (100%-loss rate). Then the bars for the plasmids which are more stable would be higher and for those that are not stable would be lower, which better corresponds to the data presented. I would suggest to repeat the experiment in triplicates and add the standard deviation bars.
  • Lines 171-177 from “The size and number of colonies…”. It would be good to introduce first in one sentence the methods which were used to compare the viability and the appearance of the cells (the drop test and the microscopy). Like that the reader would understand easier what kind of data are presented.
  • The quality of the photo in Fig 2A is very low. What is more, the authors mention in the text that the ΔparAB2 strain forms smaller colonies than the wt strain, and in the picture, we could not see these smaller colonies, it seems like it does not grow at all.
  • Lines 185-187 from “Taken together…”. I feel that from sentence authors should start a new paragraph, as the sentence summarize all the results presented in this section and not only the microscopy data.
  • Lines 209. If I understand well, the names of the strains are inverted here (ΔhupB ΔparAB2 ori2::2parS1 should go in place of ΔparAB2 ori2::2parS1 and opposite).
  • Lines 211-214. ‘The single R2II focus of the smaller ΔparAB2 ori2::2parS1 and ΔhubP ΔparAB2 ori2::2parS1 cells located at the pole opposite to the L1I pole whereas the single R2II focus of the smaller WT cells was positioned close to mid-cell Fig 2B).’ This sentence is a bit confusing, as the localization of the foci in ΔparAB2 ori2::2parS1 are more polar than in ΔhubP ΔparAB2 ori2::2parS1. Could it be explained better?
  • Lines 220-221. Should it not be discussed somewhere in the discussion rather than here?
  • Lines 228-229. ‘In each cell scheme, the old pole and the new pole are by a black and white shading, respectively.’ It would be enough to put this information once in the description of Fig 2. and to remove it from here.
  • Why in the Hub+ background ori1 and ori2::2parS1 are localized randomly on the old or new pole, and in the Hub- background ori1 is localized on the old pole, as in the wt strain? Should it not be discussed in the discussion?
  • Line 246. ‘for the two strains.’ I would specify which two strains, it is not easy to follow.
  • Fig 3. How does the HubP protein maintain the chr1 partition to 1/4 and 3/4 in the ΔparS1 strain (Fig 3B vs Fig 3D). Should it not be discussed in the discussion?
  • 4H does not have any description. The presented model shows the chr1 rearrangement in the ΔhubP strain, which is based on the experiments presented in the study. At the same time, it shows unchanged arrangement of chr2 in the ΔhubP strain. The arrangement of chr2 was not tested in this study. Do the authors have any data to assume that the chr2 is not changed in the ΔhubP strain? If not, chr2 should be removed from the model or it should be indicated that its arrangement is only assumed and might be different.
  • Lines 321-322. To be strict, the authors investigated only HubP, but not PopZ.
  • Lines 325-326. ‘Our data unambiguously demonstrated that the ParABS1 system of V. cholerae continues to exhibit partitioning activity in the absence of HubP.’ I would suggest to add the information that it was demonstrated in both natural and heterologous host (E. coli). Like that the results from E. coli are at least represented in the discussion.
  • Line 339-342. ‘As ParA1 is now free to be reactivated anywhere in the cell, the nucleoid could be constantly recoated of active ParA1. The parB1-parS1 complex would then be trapped in highly dense regions of the nucleoid (HDR), as proposed for plasmid systems [43].’ What do the authors mean by saying ‘now’? In ΔhubP?
